# An L-threonine transaldolase is required for L-*threo*-β-hydroxy-α-amino acid assembly during obafluorin biosynthesis

Thomas A. Scott[1], Daniel Heine[1], Zhiwei Qin[1] & Barrie Wilkinson[1]

β-Lactone natural products occur infrequently in nature but possess a variety of potent and valuable biological activities. They are commonly derived from β-hydroxy-α-amino acids, which are themselves valuable chiral building blocks for chemical synthesis and precursors to numerous important medicines. However, despite a number of excellent synthetic methods for their asymmetric synthesis, few effective enzymatic tools exist for their preparation. Here we report cloning of the biosynthetic gene cluster for the β-lactone antibiotic obafluorin and delineate its biosynthetic pathway. We identify a nonribosomal peptide synthetase with an unusual domain architecture and an L-threonine:4-nitrophenylacetaldehyde transaldolase responsible for (2S,3R)-2-amino-3-hydroxy-4-(4-nitrophenyl)butanoate biosynthesis. Phylogenetic analysis sheds light on the evolutionary origin of this rare enzyme family and identifies further gene clusters encoding L-threonine transaldolases. We also present preliminary data suggesting that L-threonine transaldolases might be useful for the preparation of L-*threo*-β-hydroxy-α-amino acids.

[1] Department of Molecular Microbiology, John Innes Centre, Norwich Research Park, Norwich NR4 7UH, UK. Correspondence and requests for materials should be addressed to B.W. (email: barrie.wilkinson@jic.ac.uk).

D espite the great structural diversity of biologically important natural product scaffolds, a number of privileged structural motifs occur within them with high frequency. An important example is found in the β-hydroxy-α-amino acids which are constituents and intermediates in the biosynthesis of many agriculturally and medicinally valuable bioactive natural products. These include the antibiotics chloramphenicol[1] and vancomycin[2], sphingofungin antifungal agents[3] and the proteasome inhibitor salinosporamide[4] among others. They are also of great importance as chiral building blocks for chemical synthesis and L-*threo*-3,4-dihydroxyphenylserine (Droxidopa) is itself a (pro)drug used in the treatment of Parkinson's disease[5]. Although many different synthetic methods have been reported for their chemical synthesis —including methods based on glycine enolates[6], glycinamides[7], glycine Schiff's base[8] and the aminohydroxylation of olefins[9]—the production of β-hydroxy-α-amino acids by enzymatic means is particularly attractive as these can set two stereocentres in a single reaction that can be performed in a one-pot process with minimal protection of substrates and under mild, aqueous conditions. One such example is found in the threonine aldolases (TAs) although their utility is limited due to low synthetic yields and modest diastereoselectivity (they are highly stereoselective for the α-carbon)[10–12]. As such the discovery and characterization of new enzymes with utility for the synthesis of β-hydroxy-α-amino acids is desirable to expand our toolkit for asymmetric synthesis. Given the presence of β-hydroxy-α-amino acid derived moieties in many natural product structures we decided to target biosynthetic gene clusters (BGCs) encoding for the production of such molecules.

On this basis the antibacterial agent obafluorin[13,14] (**1**; Fig. 1) produced by *Pseudomonas fluorescens* ATCC 39502 attracted our attention as classical feeding experiments suggested its β-lactone moiety derives from cyclization of an unusual β-hydroxy-α-amino acid intermediate (2*S*,3*R*)-2-amino-3-hydroxy-4-(4-nitrophenyl)butanoate (**2**) via homologation of L-4-aminophenylalanine (**3**)[15–18]. Previously, Herbert and Knaggs proposed a biosynthetic pathway in which **2** derives from 4-aminophenylpyruvate (**4**; derived from transamination of **3**) and glyoxylate via a decarboxylative, thiamine diphosphate (ThDP)-dependent mechanism as shown in Supplementary Fig. 1 (ref. 16). If correct, such an enzyme, when coupled with appropriate aminotransferases, would provide access to valuable substituted β-hydroxy-α-amino or α-hydroxy-β-amino acids in enantiomerically enriched form.

Moreover, the biosynthesis of **1** is of additional interest as the β-lactone structural motif occurs infrequently in nature but is usually associated with molecules possessing potent and valuable biological activity[19,20]. For example, **1** exhibits narrow spectrum antibacterial activity[13,14], despite being a substrate for β-lactamases and sensitive to acidic conditions, undergoing facile hydrolysis or ring opening in the presence of nucleophiles. It protects mice infected with a clinical isolate of *Streptococcus pyogenes* when dosed systemically, and microscopic examination of *Escherichia coli* grown at sub-lethal doses showed unusual cell elongation[14]. Taken together these observations suggest that **1** acts in a specific manner rather than as a general acylating agent (β-lactones are generally effective electrophiles able to form reversible covalent linkages with nucleophilic residues of target proteins). The molecular target of **1** remains elusive making studies on its biological activity and biosynthesis of particular interest: in an era of extensive antimicrobial resistance the identification of new antibacterial targets is of great potential significance.

Using genome mining, in conjunction with mutational and biochemical analysis, we report the pathway for the biosynthesis of **1** and show that the BGC includes genes involved in the biosynthesis of the precursors **4** and 2,3-dihydroxybenzoic acid (**5**). These co-localize with additional genes including those encoding for a ThDP-dependent phenylpyruvate decarboxylase (pPDC) ObaH and a putative serine hydroxymethyltransferase (SHMT)/L-TA ObaG, both of which are essential for the biosynthesis of **2** and therefore, ultimately, **1**. Our data show that ObaG actually encodes for an L-threonine transaldolase (L-TTA), and we propose that this group of enzymes may offer utility as tools for the synthesis of L-*threo*-β-hydroxy-α-amino acids.

## Results

**Identification of the 1 BGC.** The genome of *P. fluorescens* ATCC 39502 was sequenced at the Earlham Institute (Norwich, UK) using the Pacific Biosciences (PacBio) RSII platform, and assembly using the HGAP2 pipeline gave a single circular contig of ∼6.15 Mb. The sequence was submitted to the open access genome mining platform antiSMASH 3.0 (ref. 21) which predicted sixteen BGCs, seven of which included nonribosomal peptide synthetase (NRPS) encoding genes. Functional assignments were made for each gene by comparing the deduced amino acid sequence with proteins of known function which allowed us to identify one BGC encoding the activities we anticipated would be required for the biosynthesis of **1** (Fig. 1; Supplementary Table 1). The *oba* BGC sequence has been deposited with GenBank under the accession no. KX931446.

To verify the *oba* locus we subjected the BGC to mutational analysis (see below). To facilitate genetic modification we generated pTS1 (Supplementary Fig. 2 and Supplementary Table 2; GenBank accession no. KX931445), a variant of the suicide vector pME3087 (ref. 22) which was first sequenced *de novo* by primer walking (Supplementary Table 2; GenBank accession no. KX931444). pTS1 incorporates an expanded multiple cloning site and the *sacB* gene which allows for positive selection of gene replacements after secondary recombination and plasmid loss[23] (see Supplementary Note 1 for construction). Gene inactivation experiments were performed by generating in frame deletions using pTS1, and all resulting mutants were verified by PCR amplification across the deleted genomic region followed by sequencing of the amplicon; at least three independent mutants were checked. All mutants were functionally confirmed and checked for the lack of polar effects by genetic complementation through ectopic expression of the deleted gene under the control of the *tac* promoter using the expression plasmid pJH10TS, based on pJH10 previously reported by Thomas and co-workers[24] (Supplementary Note 2 and Supplementary Table 2). As required, chemical complementation with putative pathway intermediates was carried out in parallel by the addition of exogenous material to growing cultures. All strains and their complemented derivatives were grown in triplicate under **1** producing conditions (alongside the wild type (WT) strain), extracted with ethyl acetate and the extracts subjected to high performance liquid chromatography (HPLC) and high performance liquid chromatography-mass spectrometry (HPLC-MS) analysis.

**Bioinformatics and mutational analysis of the 1 BGC.** Bioinformatics analysis suggests that three genes (*obaJLN*) encode for the well understood pathway to **5** from chorismate[25,26] and deletion of either *obaJ* (an isochorismatase) (Supplementary Fig. 3) or *obaL* (a 2,3-dihydro-2,3-dihydroxybenzoate dehydrogenase) (Fig. 2a) led to the loss of **1** production and 4 to 5-fold elevated levels of 4-nitrophenylethanol (**6**) and 4-nitrophenylacetic acid (**7**), which are proposed shunt metabolites of the biosynthesis of **2** (ref. 17). The addition of exogenous **5** was able to restore the biosynthesis of **1** (to 71% (Δ*obaJ*) and 68% (Δ*obaL*) of WT titres), as was

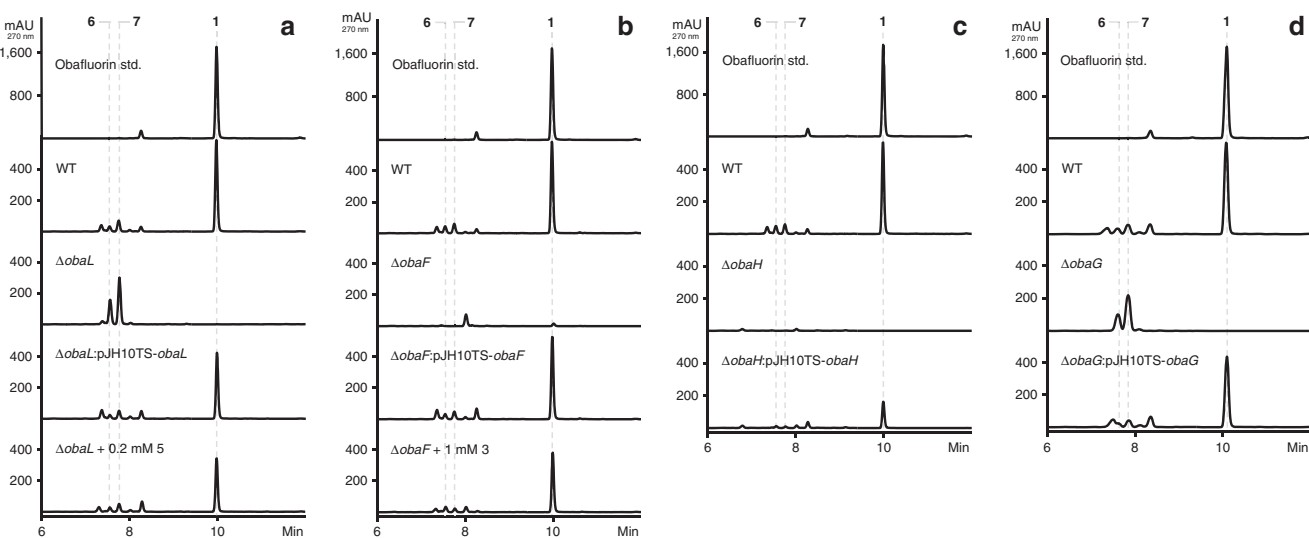

**Figure 1 | Gene cluster architecture and proposed biosynthetic pathway for obafluorin biosynthesis in *P. fluorescens* ATCC 39502.** Obafluorin protein coding sequences *obaA-N* are labelled and coloured according to related biosynthetic functions, with operons indicated with black arrows beneath the gene cluster. The domain architectures of the aryl carrier protein ObaK and nonribosomal peptide synthetase ObaI are represented (ArCP = Aryl carrier protein; C = Condensation domain; A = Adenylation domain; PCP = Peptidyl carrier protein domain; TE = Thioesterase domain; MbtH = MbtH-like domain), and the anticipated mechanism of TE-catalysed peptide release and β-lactone ring closure is also illustrated. **6** and **7** represent breakdown products of **10**, and were found to accumulate when either the biosynthesis of **5**, or the assembly of **1**, was disrupted. Compounds **3** and **8** were used in chemical feeding experiments in this work, but were not shown to be intermediates in the biosynthesis of **1**.

**Figure 2 | HPLC profiles for mutagenesis and complementation experiments in selected *oba* genes.** (**a**) *obaL*, (**b**) *obaF*, (**c**) *obaH* and (**d**) *obaG*. Numbered peaks refer to key products and shunt metabolites identified in Fig. 1.

ectopic expression of the deleted gene (to 73% (*obaJ*) and 85% (*obaL*) of WT titres). Moreover, by blocking production of the key intermediate **5** (and therefore the pathway endpoint **1**), we anticipated that the predicted pathway intermediate **2** (refs 16,17) might accumulate (see Fig. 1). However, after exhaustive LCMS analysis we could not observe any accumulation of **2**, nor of the potentially related pathway intermediate (2-amino-3-hydroxy-4-(4-aminophenyl)butanoate (**8**). Synthetic standards of both **2** and **8** were prepared to aid this analysis (see Materials and Methods).

The three genes (*obaDEF*) appear sufficient to encode for the production of **4**, the biosynthesis of which is well understood from previous genetic studies on chloramphenicol[27] and pristinamycin[28] biosynthesis. Consistent with these predictions, deletion of the bifunctional gene *obaF* (a bifunctional 4-aminochorismate mutase/4-aminoprephenate dehydrogenase) led to the loss of **1**, **6** and **7** production (Fig. 2b). Addition of exogenous **3** was sufficient to restore efficient production of **1** (between 68%-71% of WT titres), as was the ectopic expression of *obaF* (between 73 and 90% of WT titres). We presume that **3** is converted to **4** by transamination *in vivo*. Surprisingly, no accumulation of the pathway intermediate **5** could be observed in the *obaF* mutant.

A putative type II 3-deoxy-D-arabino-heptulosonate-7-phosphate (DAHP) synthase is encoded by *obaM*. Homologues of this gene have been observed in a number of bacterial BGCs[4,27,29], where their products are believed to mitigate the aromatic amino acid-based negative feedback of primary metabolic DAHP synthases and drive flux into specialized metabolism. This indicated a likely role for ObaM in chorismate supply for the biosynthesis of both **2** and **5**. Consistent with this deletion of *obaM* led to the almost complete abolition of **1** biosynthesis (Supplementary Fig. 4), but we were surprised to observe the accumulation of **6** and **7**. Moreover, exogenous supply of **5** alone was sufficient to restore production of **1** (68–72% of WT titres), further indicating that biosynthesis of **2** is unperturbed by *obaM* deletion, which appears instead to only impact **5** biosynthesis. Consistent with this, exogenous supply of **3** alone could not recover **1** production but did lead to further elevated levels (five- to sevenfold) of **6** and **7**. Ectopic expression of *obaM* recovered **1** production (50–55% of WT titres), precluding the possibility of polar effects on *obaJLN*.

The gene *obaC* encodes a putative non-heme di-iron mono-oxygenase related to AurF from the aureothin pathway (73% query cover, 25% identity)[30,31] and ClmL from the chloramphenicol pathway (72% query cover, 19% identity)[32,33]. Both enzymes are arylamine oxygenases which catalyse the conversion of an aromatic amine to a nitro group, and we propose ObaC catalyses the conversion of **4** to 4-nitrophenylpyruvate (**9**). Consistent with this role, deletion of *obaC* abolished production of **1**, **6** and **7** (Supplementary Fig. 5). The biosynthesis of all three compounds was re-established after ectopic complementation with *obaC* (81% of WT titres of **1**). Surprisingly, only the addition of exogenous synthetic **9** to growing cultures of this mutant was able to restore **1** biosynthesis (to 29% of WT titres), whereas addition of **2** did not. We believe this is likely due to failure of this metabolite to penetrate the *Pseudomonas* cell membrane based on subsequent data below.

At the centre of the BGC is the bimodular NRPS-encoding gene *obaI*, the product of which is predicted to catalyse formation of the amide bond between **2** and **5**. The thioesterase (TE) domain of ObaI is unusually located between the peptide carrier protein (PCP$_2$) domain and first adenylation (A$_1$) domain, and we predict that it catalyses release of the resulting enzyme bound *pseudo*-dipeptide with concomitant formation of the β-lactone moiety to yield **1**. Analysis of the A domain sequences of ObaI using NRPSpredictor2 (ref. 34) identified **5** as the likely substrate of A$_1$, whereas L-threonine is predicted to be the

substrate of A$_2$. This means that the domain architecture of ObaI (Fig. 1) is further unusual as the putative A$_1$ domain is located at the C terminus of the enzyme, rather than towards the N terminus as would be expected. Moreover, A$_1$ is adjacent to an embedded MbtH-like protein domain. MbtH-like proteins are auxiliary proteins required for the activity of many, but not all NRPS A domains[35–37]. Some A domains require MbtH-like proteins for their functional expression, suggesting they may have a chaperone-like function[38]. Discrete MbtH-like proteins are the most commonly observed, but the identification of MbtH-like domains embedded within an NRPS, as for ObaI here, is rare and has been reported for only two other BGCs, those responsible for the biosynthesis of nikkomycin[39] and streptolydigin[40]. While the amino acid residues critical for intra-domain interaction have been well characterized in the latter case, no biochemical data exists to verify that either of these *in cis* domains are functionally essential *in vivo*, and the exact role of MbtH-like proteins in NRPS catalysis is yet to be determined. Deletion of *obaI* abolished production of **1**, which was recovered to 79% of WT levels on ectopic complementation (Supplementary Fig. 6). The Δ*obaI* mutant accumulated the shunt metabolites **6** and **7** at similar levels to the WT strain, but, once again, exhaustive analysis of the fermentation extracts did not show any accumulation of **2** or **5**, the putative substrates of ObaI.

ObaI contains only a single PCP$_2$ domain where two such domains would be expected for its correct function. Further analysis of the BGC suggested that *obaK* encodes a discrete aryl carrier protein (ArCP$_1$). The presence of an ArCP is anticipated as this protein function is required for the tethering of **5** (following activation by the A$_1$ domain) to participate in amide bond formation (Fig. 1). On the basis of these analyses we hypothesize that the unusual positioning of A domains within the assembly line provides access for both to the embedded MbtH-like domain, a likely prerequisite for efficient catalysis. Given the canonical structure and function of NRPS assembly line enzymes, this organization will preclude functional interaction of the A$_1$ domain and the condensation (C$_2$) domain, a consequence of which is the requirement for a discrete, and therefore mobile, ArCP$_1$ (ObaK) to enable both interaction with the A$_1$ domain for acylation with **5** and subsequent interaction with the C$_2$ domain for peptide bond formation.

Deletion of *obaK* also abolished production of **1**, which was recovered by ectopic expression (to 77% of the WT level) (Supplementary Fig. 7). The titres of **6** and **7** also increased significantly (approximately fourfold) but again neither **2** nor **5** was accumulated. Exogenous supply of **5** did not lead to production of **1** confirming that this phenotype is not linked to a lack of **5** biosynthesis. We queried this possibility as *obaK* appears to have evolved by the splitting of an ancestral *entB*-like gene, which usually encode for bidomain proteins involved in the biosynthesis of **5** (refs 25,26). Indeed, the adjacent *obaJ* gene corresponds to the N-terminal isochorismatase domain of EntB and is required for production of **5** during biosynthesis of **1**. The biosynthesis of **5** does not require the ArCP domain of EntB[41] consistent with our various results and biosynthetic hypothesis.

Our analysis thus left the biosynthesis of **2** as the remaining unknown, and only the products of *obaG* and *obaH* to be functionally assigned. *obaG* encodes a putative SHMT/L-TA while *obaH* encodes a putative ThDP-dependent pPDC. Mutation of both genes abolished the production of **1**, but the accumulation of **6** and **7** was only observed in the *obaG* mutant (Fig. 2c,d). Again, exogenous addition of **2** did not recover **1** production in these mutants whereas ectopic expression of the deleted genes did (to 62% (*obaG*) and 29% (*obaH*) of WT titres). These mutational data are consistent with the role of ObaG as the enzyme directly responsible for **2** production as proposed in Fig. 1, but did not

provide direct support for **2** as a biosynthetic intermediate. In contrast, the accumulation pattern for the shunt metabolites **6** and **7** strongly suggested that 4-nitrophenylacetaldehyde (**10**) might be a key intermediate. This reactive and potentially toxic molecule is likely to be catabolized to the shunt metabolites **6** and **7** via standard detoxification pathways before accumulating to any degree. Consistent with this hypothesis, addition of exogenous **10** to growing cultures of the ΔobaG mutant did not restore **1** production but did lead to elevated levels of **6** and **7**. We also noted that chemical complementation of mutants deficient in **5** biosynthesis are sensitive to the concentration of exogenous material added. Production of **1** was re-established when **5** was added to give a final concentration of 0.2 mM, but at higher concentrations (for example, ≥0.5 mM) both growth of *P. fluorescens* and biosynthesis of **1** were significantly affected. As noted above we could not observe the presence of **2** or **5** in any of the mutants anticipated to accumulate them. Additionally, in mutants where **2** or its aldehyde precursor **10** might have accumulated, we always observed elevated levels of the shunt metabolites **6** and **7**, breakdown products of aldehyde **10**, a highly reactive and toxic molecule. Taken together these combined data suggest that the biosynthetic pathway is tightly regulated to avoid the accumulation of intermediates deleterious to cell growth and survival. Interestingly, the intermediary of free phenylacetaldehyde, a reactive aldehyde structurally similar to **10**, is avoided in the recently reported ripostatin biosynthetic pathway via the activity of a pyruvate dehydrogenase-like complex which functions to decarboxylate phenylpyruvate and then retain the phenylacetaldehyde product as a phenylacetyl-S-carrier protein species[42]. Finally, bioinformatics suggests the obafluorin BGC is regulated by quorum sensing under the control of the *luxIR* homologues *obaAB*.

**2 is assembled by sequential activity of ObaH and ObaG.** The essential nature of *obaG* and *obaH*, in conjunction with the accumulation pattern of **6** and **7** by their mutants, led us hypothesize that the biosynthesis of **2** involves ThDP-dependent decarboxylation of **9** by ObaH to yield the aldehyde **10**. This would then be used as a substrate by the pyridoxal-5′-phosphate (PLP) dependant enzyme ObaG acting as an aldolase to give **2**. Both enzymes were readily expressed as soluble proteins in hexahistidine-tagged form using *E. coli* NiCo21(DE3) pLysS. They were purified using standard procedures and their identity verified by tandem MS.

Due to the original proposal of Herbert and Knaggs that a pPDC-like enzyme might produce an acyloin intermediate (Supplementary Fig. 1)[16] we first tested ObaH with **9** plus glyoxylate as substrates using a discontinuous format coupled to independent LCMS/MS and HPLC-UV assays. In none of these reactions could we detect any acyloin products whereas the ketoacid substrate was depleted and aldehyde **10** accumulated whenever ObaH was incubated with **9**, clearly identifying it as a **9** decarboxylase (Fig. 3a). To examine the ability of the putative L-TA-like enzyme ObaG to produce **2**, it was incubated with varying concentrations of **10** and glycine using the same assay format as for ObaH. Under none of these conditions could we observe production of **2**. At this point we considered the possibility that ObaG might actually be a transaldolase and incubated it with **10** and either L-serine or L-threonine instead of glycine, and were gratified when we observed excellent production of the anticipated intermediate **2** only when L-threonine was present in the reaction mixture (Fig. 3b). Moreover, when [U-$^{13}$C$_4$,$^{15}$N$_1$]L-threonine was used as substrate we were able to demonstrate, by $^{13}$C NMR spectroscopy and high-resolution (HR)-LCMS/MS, the regiospecific transfer

of three heavy isotope atoms to the appropriate positions in the product **2**, in addition to the formation of [1,2-$^{13}$C$_2$]acetaldehyde (Figs 3c and 4). The structure and stereochemistry of isolated **2** was then verified by comparison of its NMR spectra and optical rotation to those of previously synthesized material[43]. To confirm the biosynthetic relevance of L-threonine *in vivo* we fed [U-$^{13}$C$_4$,$^{15}$N$_1$]L-threonine to growing cultures of the WT producer and examined the resulting **1** by HR-LCMS/MS analysis. This indicated the highly efficient, site specific incorporation of a $^{13}$C$_2^{15}$N$_1$ unit as anticipated (Supplementary Fig. 8). In a final experiment, we were able to couple the ObaH decarboxylase and ObaG L-threonine transaldolase (L-TTA) reactions *in vitro* (Fig. 3c,d), to catalyse the formation of **2** using L-threonine and **9** as the only substrates. The proposed mechanism for the biosynthesis of **2** by ObaG is shown in Supplementary Fig. 9.

Having characterized the substrates and function of both ObaH and ObaG, we sought to further characterize ObaG by confirming its identity as a PLP-dependent enzyme and attempted to collect preliminary kinetic data. Despite being identified as a putative SHMT, ObaG shares relatively little sequence similarity to *bona fide* bacterial SHMTs (25% amino acid sequence identity), but does comprise several residues crucial for PLP-dependent activity, most importantly the lysine residue required for internal aldimine formation (Supplementary Fig. 10). ObaG displays a UV/Vis spectrum (Supplementary Figs 11a and b) characteristic of PLP-dependent proteins[44], comprising absorption maxima at 340 and 390 nm that correspond to the equilibrium established between the enolimine and ketoenamine forms of PLP respectively. ObaG is unusual in that it exhibits a distinct salmon pink colour in solution (Supplementary Fig. 11c), which creates an additional absorbance maximum at 512 nm (Supplementary Figs 11a and b). Treatment of ObaG in the absence of excess PLP with 10 mM L-penicillamine led to the anticipated formation of a thiazolidine adduct (Supplementary Fig. 11d) determined by UV absorbance at 340 nm (Supplementary Fig. 11a), with concomitant loss of the ketoenamine peak at 390 nm, as observed previously by Lowther *et al.*[44] for serine palmitoyltransferase. ObaG was further incubated with NaBH$_4$ (1 mM) resulting in the formation of a peak at 330 nm consistent with the formation of the reduced ObaG-PLP amine adduct[45] (Supplementary Figs 11b and e). In both experiments a decrease in the 512 nm peak was also observed. Taken together these results strongly support ObaG being a PLP-dependent enzyme.

Having characterized the L-TTA reaction catalysed by ObaG, and its PLP-dependence, we acquired preliminary kinetic data. A time course was determined using standard L-TTA assay conditions showing that equilibrium between substrate **10** and product **2** was achieved within ~1 h after reaction initiation (Supplementary Fig. 12a). Unfortunately, the UV/Vis spectrum of **10** precluded the use of a continuous coupled assay based on acetaldehyde accumulation due to overlap with that of the measured side-product, and its inherent reactivity confounded discontinuous methods measuring product formation with varying **10**. The best results were achieved by varying L-threonine concentration using a discontinuous HPLC-based approach. Single-substrate kinetic analysis of His$_6$-ObaG revealed typical Michaelis–Menten kinetics with respect to varying L-threonine, yielding kinetic constants of $K_m = 40.2 \pm 3.8$ mM and $k_{cat} = 62.9 \pm 1.9$ min$^{-1}$ (Supplementary Fig. 12b).

**Preliminary investigation of the NRPS ObaI.** In a final biochemical experiment we were able to express the NRPS ObaI as the full length (211.5 kDa) protein in its *apo*-form and probe the ability of its two A domains to activate various amino

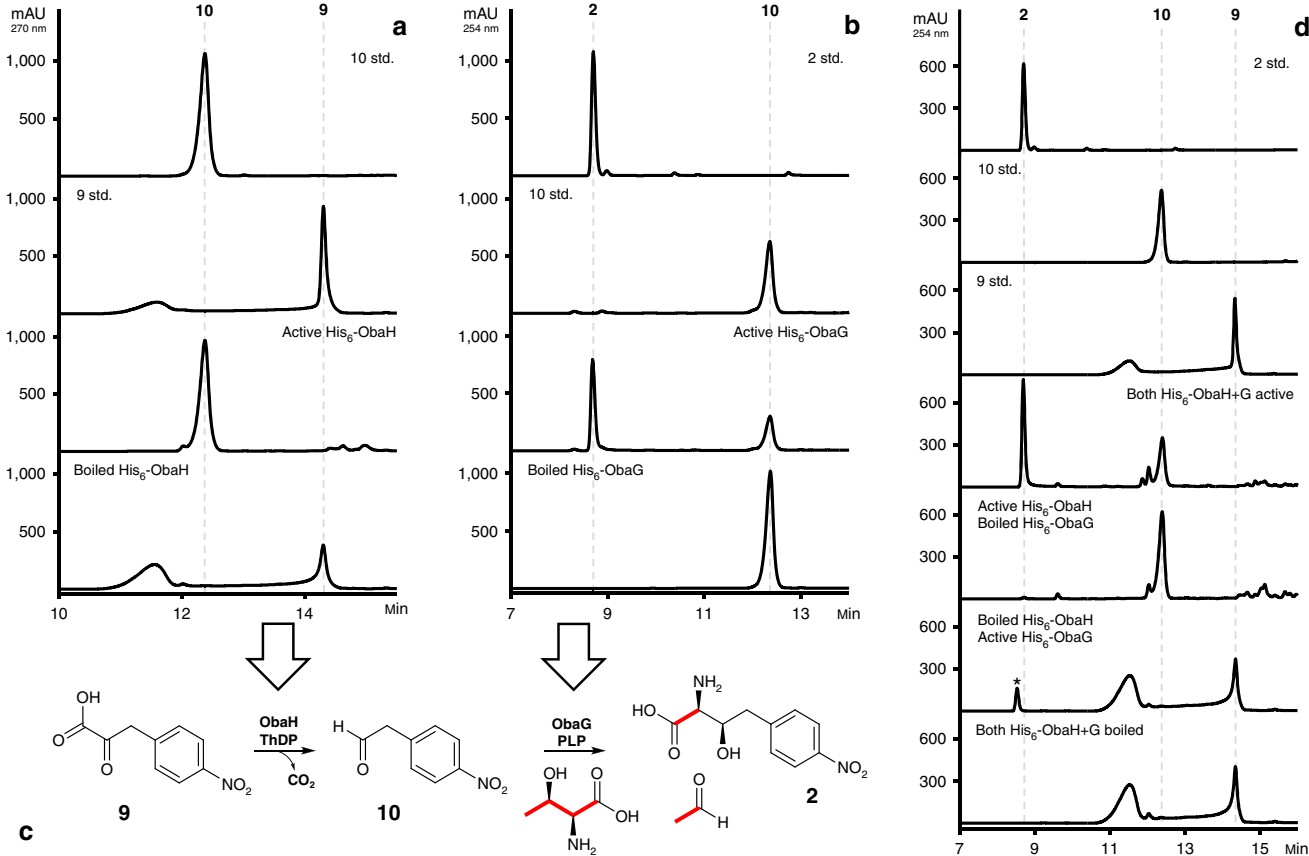

**Figure 3 | Experimental characterization of the biosynthesis of 2.** (**a**) His$_6$-ObaH decarboxylase activity assay HPLC profiles. (**b**) His$_6$-ObaG L-TTA activity assay HPLC profiles. (**c**) Illustration of the biosynthesis of **2** in a two-step reaction catalysed by consecutive action of ObaG and H. The His$_6$-ObaG L-TTA activity assay was also performed using [U-$^{13}$C$_4$,$^{15}$N$_1$] L-threonine, and terminated reactions were purified and analysed by NMR (Fig. 4) to show $^{13}$C incorporation into **2**. The result of this experiment is also illustrated here. (**d**) Coupled decarboxylase and L-TTA assay HPLC profiles. **9** was also found to be a poor substrate for the ObaG-catalysed L-TTA reaction, yielding unknown product*. In each case substrate/product peaks are numbered in accordance with Fig. 1.

acids using a discontinuous hydroxamate formation assay[46] (Methods section). While this approach did not allow us to dissect the activities of the two A domains independently, it did suggest that **2** is the preferred substrate for the A$_2$ domain, but that it displays relaxed substrate specificity with activation of L-tryptophan and L-serine being observed (Supplementary Fig. 13). It should be noted that NRPSpredictor2 (ref. 34) identified L-threonine as the substrate for the A$_2$ domain. The A$_1$ domain, which is predicted to activate **5**, also appears to have relaxed specificity as benzoic acid was activated in addition to **5**. Despite the apparent flexibility of both A domains we could not observe the production of obafluorin congeners in the WT strain, although such molecules might be sensitive to proteolytic degradation. Most likely, and consistent with the observations of others, while the A domains of intact ObaI show an inherent substrate flexibility, the C$_2$ domain is likely to exert a gatekeeper function with regards to the selection of substrates for carrier protein acylation and subsequent amide bond formation[47].

**Phylogenetic analysis of L-TTAs.** L-TTA activity like that of ObaG here has previously been described only in the biosynthesis of 4-fluoro-L-threonine[48] and in 5′-C-glycyluridine (GlyU), a key constituent of lipopeptidyl nucleoside natural products[49] (Supplementary Fig. 14). Both 4-fluoro-L-threonine transaldolase

(FTase) and L-threonine:uridine-5′ transaldolases of the LipK family were all identified as putative SHMTs, but have been shown to catalyse L-threonine β-substitutions via the sequential breaking and forming of Cα − Cβ bonds. We have now undertaken detailed phylogenetic analyses of PLP-dependent SHMTs, L-TAs and the L-TTAs described here (Fig. 5)[50]. This reveals that bacterial L-TTAs and SHMTs diverged from each other to form discrete clades, and that they and the TAs diverged from an earlier common ancestor. It is particularly notable that the common ancestor appears to have diverged into enzymes that all utilize L-threonine, other than the SHMTs. While the evolutionary relationship between ObaG and FTase is not so easily determined when comparing different phylogenies (Supplementary Figs 15 and 16), what is clear is that they are distinct from the L-threonine:uridine-5′ transaldolase (LipK) clade. In addition, OrfA—identified as a glycine hydroxymethyltransferase and proposed as a biosynthetic enzyme required for the biosynthesis of alanylclavam in the cephamycin/clavulanic acid/5S-clavam producer *Streptomyces clavuliguris*[51]—also consistently grouped within the L-TTA clade. The biochemical characterization of this enzyme has not yet been reported, but based on our analysis we suspect that it has been incorrectly annotated and is in fact an L-TTA using L-threonine as a substrate rather than an L-TA using glycine as previously proposed (Supplementary Fig. 14).

## Discussion

On the basis of our bioinformatics analysis combined with mutational, biochemical and chemical feeding results we propose the biosynthetic pathway for **1** shown in Fig. 1. The pathway involves a unique, non-canonical NRPS which carries an integrated MbtH-like protein domain and contains a central TE domain which is likely to be responsible for β-lactone formation. Central to the pathway, however, is a new route for

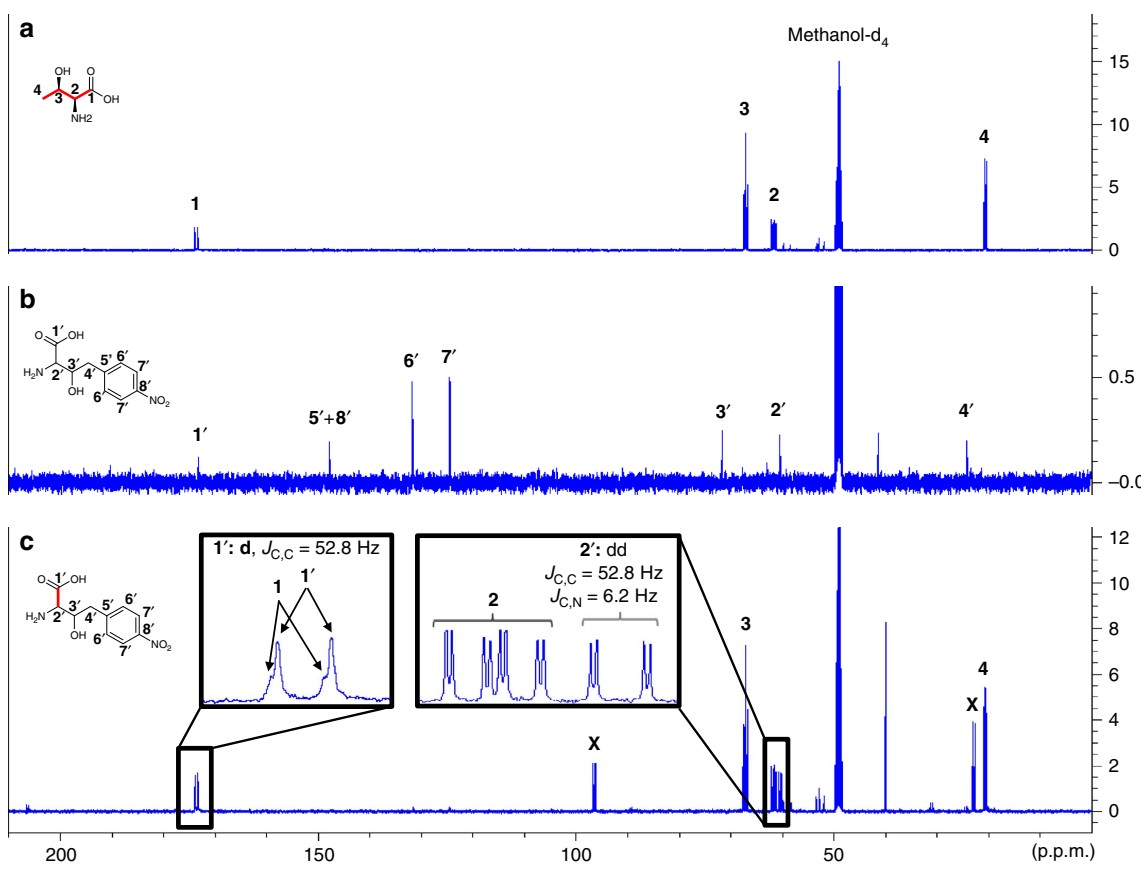

**Figure 4 | Characterization of the biosynthesis of 2 by ObaG using $^{13}$C NMR.** (**a**) $^{13}$C NMR spectrum of [U-$^{13}$C$_4$,$^{15}$N] L-threonine in methanol-d$_4$. (**b**) $^{13}$C NMR spectrum of the synthetic reference **2**. (**c**) $^{13}$C NMR spectrum of enzymatically synthesized **10**. Signals marked with **X** correspond to the two acetal forms of acetaldehyde and glycerol, being formed after release of acetaldehyde and therefore proving threonine L-TTA activtity of ObaG.

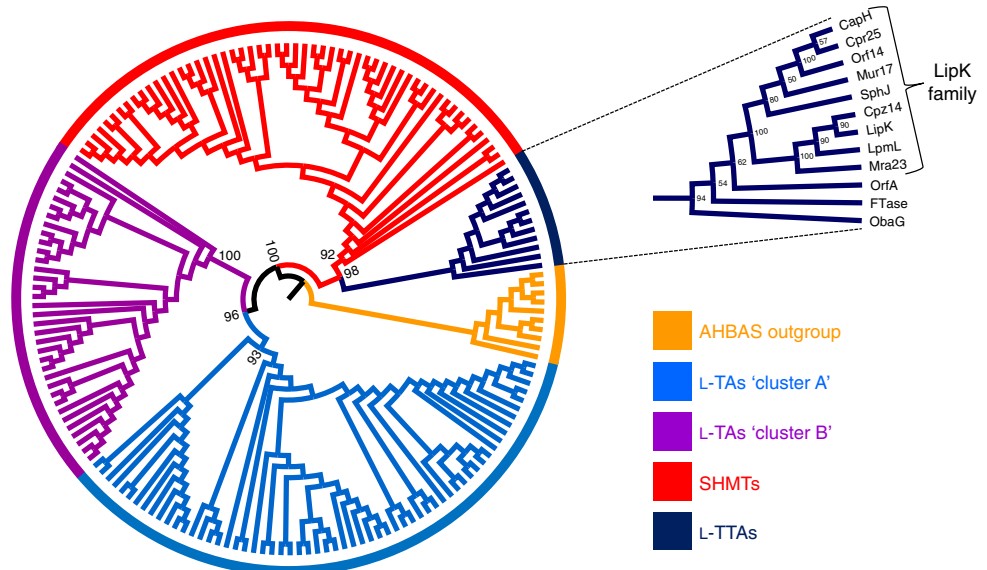

**Figure 5 | Maximum likelihood tree of L-TAs, SHMTs and L-TTAs.** A set of 3-amino-5-hydroxybenzoate synthase (AHBAS) amino acid sequences serve as the outgroup. RAxML likelihood values at the root nodes for SHMT and L-TA cluster clades are annotated. The clade comprising the L-TTA ObaG characterized in this work has been expanded and annotated with RAxML likelihood values and the identity of amino acid sequences represented.

biosynthesis of the homologated L-*threo*-β-hydroxy-α-amino acid **2** via sequential decarboxylation and transaldolase reactions. L-TTA reactions are rare and have been reported previously only for the biosynthesis of 4-fluoro-L-threonine and GlyU from fluoroacetaldehyde[48] and uridine-5′-aldehyde[49], respectively (Supplementary Fig. 14), both of which exhibit the L-*threo* stereochemistry of **2**. Detailed phylogenetic analysis has shed light on the evolutionary origins of the L-TA (which use glycine as substrate) and L-TTA families (Fig. 5), and suggests that the two families diverged from a common ancestor with the transaldolase family subsequently splitting again into lineages that specifically utilize either L-threonine (L-TTAs) or L-serine (SHMTs) as substrates. With this knowledge in hand we used a phylogeny-based, mechanism-guided genome mining approach to identify BGCs containing L-TTAs, and therefore biosynthetic pathways likely involving a β-hydroxy-α-amino acid intermediate. This led to the identification of a putative L-TTA involved in the biosynthesis of a 5S-clavam metabolite, indicating phylogenetic analysis as a potentially general genome mining approach for the identification of further L-TTAs.

We also identified multiple BGCs likely to encode for the production of **1** or very closely related compounds in the genomes of several *Pseudomonas* environmental isolates[52], *Burkholderia* strains (Genome sequences for *Burkholderia* species containing *oba* like BGCs: *Burkholderia stagnalis* (NZ_LPGD01000044.1); *Burkholderia diffusa* (NZ_LOTC01000033.1), *Burkholderia territorii* (NZ_LOSY01000044.1), *Burkholderia ubonensis* (NZ_CP013463.1).) and a *Chitiniphilus* strain (Genome sequences for *Chitiniphilus* species containing *oba* like BGCs (GenBank Accession: NZ_KB895358.1). The relatively common occurrence of the **1** BGC is consistent with the original report describing its discovery which found that numerous *Pseudomonas* isolates, from a range of geographical locations, were able to produce **1**[13]. In several of the sequenced genomes, but not all, a set of genes encoding a glycine cleavage system[53] is located adjacent to the putative biosynthetic genes. We also identified an apparently intact glycine cleavage system (Supplementary Table 1) adjacent to the BGC for **1**.

The utility of L-TAs for the enzymatic synthesis of L-*threo*-β-hydroxy-α-amino acids from an aldehyde and glycine has been extensively investigated, but is limited due to low synthetic yields and modest diastereoselectivity[10,11]. The low yields are in part due to the reversible nature of the reaction (retroaldol cleavage of L-*threo*-β-hydroxy-α-amino acids) and an equilibrium which favours aldehyde and glycine. Due to this synthetic reactions are commonly run in the presence of an excess of glycine to shift the equilibrium to the aldol side. The highly efficient reversible nature of this reaction has, however, been harnessed for the resolution of racemic mixtures to provide D-*threo*-isomers[54].

In contrast ObaG gives **2** as a single stereoisomer in an excellent 55–59% yield based on accumulation of **2** (Supplementary Fig. 12a) without the need for an excess of L-threonine. On the basis of this result we undertook preliminary investigations into the ability of ObaG to accept the alternative aldehyde substrates phenylacetaldehyde and benzaldehyde (Supplementary Figs 17a and b). Incubation with phenylacetaldehyde led to a single product in excellent yield (∼45%, based on consumption of starting material) whose LCMS profile was consistent with the expected product (2S)-amino-(3R)-hydroxy-4-phenylbutanoate. The reaction with benzaldehyde was less efficient leading to two products in a 1:2 ratio and poor overall yield (<20%). The LCMS profile of these was consistent with the production of both L-*threo* and L-*erythro*-phenylserine, respectively (an authentic standard of the *threo* diastereoisomer was utilized). These data suggest that L-TTAs may offer an alternative to L-TAs for the synthesis of L-*threo*-β-hydroxy-α-amino acids. The equilibrium appears to lie in the favour of the aldol reaction and might be improved further

through the use of excess L-threonine and the addition of glycerol which reacts preferentially with acetaldehyde, shifting the equilibrium further in the direction of the target molecule. This is consistent with our observation that ObaG is a poor catalyst for the reverse transaldol reaction when incubated with acetaldehyde and racemic **2** made synthetically (Supplementary Fig. 17c). Moreover, our preliminary data suggest that there is a potential for high diastereoselectivity depending on the substrate aldehyde which is utilized. Future work in our lab will include additional biochemical and kinetic analysis of ObaG, and a more comprehensive investigation of its potential utility for the synthesis of enantiomerically pure L-*threo*-β-hydroxy-α-amino acids.

In summary, we have delineated the entire biosynthetic pathway to **1**, and identified a distinct enzyme family for the synthesis of β-hydroxy-α-amino acids, privileged chiral building blocks in a variety of pharmacologically and agriculturally important natural products and medicines. L-TTAs perform a unique transformation, incorporating two stereocentres in one enzymatic step, with high diastereoselectivity and excellent yields. Thorough phylogenetic investigation revealed that the L-TTAs form their own discrete evolutionary lineage, distinct from L-TAs and SHMTs, shedding light on their own evolutionary history.

## Materials

**General.** Reagents and chemicals were purchased from Alfa-Aesar, Sigma-Aldrich, Santa Cruz Biotechnology, Inc., Amatek Chemical Co., Ltd., and BD Biosciences, and were used without further purification. All strains used in this study are listed in Supplementary Table 2. *Pseudomonas fluorescens* ATCC 39502 was purchased from the American Type Culture Collection (ATCC). All strains were maintained on solid Luria-Bertani medium (with appropriate selection) at 37 °C (*E. coli*) or 28 °C (*P. fluorescens*). All solvents used for HPLC were obtained from Fisher Scientific at least of HPLC grade and were filtered before use.

**Instrumentation.** NMR spectra were recorded on a Bruker AVANCE III 400 MHz spectrometer. Chemical shifts are reported in parts per million (p.p.m.) relative to the solvent residual peak of chloroform-$d_1$ ($^1$H: 7.24 p.p.m., singlet; $^{13}$C: 77.00 p.p.m., triplet), methanol-$d_4$ ($^1$H: 3.30 p.p.m., quintet; $^{13}$C: 49.00 p.p.m., septet) or water-$d_2$ (1H: 4.79 p.p.m.). Semi-preparative and analytical HPLC was performed using an 1100 system (Agilent Technologies). Extracted samples from mutagenesis and complementation experiments were analysed using a Gemini 3 μm NX-C18 110 Å, 150 × 4.6 mm column (Phenomenex) with a gradient elution: MeCN/0.1% (*v*/*v*) TFA (H$_2$O) gradient from 10/90 to 100/0 0–15 min, 100/0 for 15–16 min, gradient to 10/90 16–16.50 min and 10/90 for 16.50–23 min; flow rate 1 ml min$^{-1}$; injection volume 10 μl. Biochemical assays were analysed using a Synergi 4 μm Fusion-RP 80 Å LC column 250 × 10 mm with a gradient elution: MeOH/0.1% (*v*/*v*) TFA (H$_2$O) gradient from 10/90 to 100/0 0–14 min, 100/0 for 14–18 min, gradient to 10/90 18–18.50 min and 10/90 for 18.50–23 min; flow rate 1 ml min$^{-1}$; injection volume 5 μl (15 μl for the reverse reaction experiments). UPLC-MS measurements were performed on a Nexera X2 liquid chromatograph (LC-30AD) LCMS system (Shimadzu) connected to an autosampler (SIL-30AC), a Prominence column oven (CTO-20AC) and a Prominence photo diode array detector (SPD-M20A). A Kinetex 2.6 μm C18 100 Å, 100 × 2.1 mm column (Phenomenex) was used for LCMS. The UPLC-System was connected with a LCMS-IT-TOF Liquid Chromatograph mass spectrometer (Shimadzu). Solid phase extraction was carried out using Discovery DSC-18 SPE Tubes, filled with 1,000 mg of octadecyl-modified, endcapped silica gel (Supelco). The specific optical rotation of compounds was measured with a Model 341 Polarimeter (PerkinElmer, Inc.).

**Cloning.** All primers used in this study are reported in Supplementary Table 3. Δ*oba* strains were generated using the suicide vector pTS1, constructed as part of this work (Supplementary Note 1). Primers were designed to amplify 800–1,200 bp flanking regions of a selected *oba* protein coding sequences (PCSs) for cloning into pTS1 between *Xba*I and *Avr*II sites, and *Avr*II and *Bmt*I sites. Flanking regions were designed to comprise 10-50 PCS codons at either end of the *oba* gene to be deleted to minimize polar effects, leaving a truncated chromosomal copy of the gene with an in-frame deletion and internal *Avr*II site cloning artefact following double homologous recombination. For genetic complementation, WT copies of *oba* PCSs were cloned from start to stop as *Bmt*I-*Kpn*I/*Xba*I fragments into pJH10TS (Supplementary Note 2) for introduction and ectopic expression in the relevant mutant strains. For protein purification and expression, WT PCSs for *obaG-I* were cloned as *Nde*I–*Xho*I fragments into pET28a(+) for expression in *E. coli* NiCo21(DE3) (NEB) pLysS.

**Mutagenesis and complementation experiments.** pTS1 knockout constructs were introduced into *P. fluorescens* ATCC 39502 via conjugation from *E. coli* S17-1 λpir and single-crossover mutants were selected for on LB Tc[25]. Positive colonies were cultured overnight in antibiotic-free medium to allow time for a second cross-over event to occur. *sacB* counter-selection could then be performed by plating culture dilutions on 10% sucrose to select against retention of the pTS1 vector backbone. Colony PCR was then performed to identify double cross-over mutants from WT colonies, and Sanger sequencing (Eurofins Genomics) was performed to confirm expected deletions. pJH10TS complementation constructs were also conjugated into the relevant Δ*oba* strains and positive clones were selected for on LB Tc[25]. Clones were screened by colony PCR and confirmed by sequencing. Δ*oba* strains were complemented chemically where possible by introduction of compounds when production cultures were established. Compounds fed and the concentrations used are described in the main text.

**Analysis of metabolite production.** WT and recombinant *P. fluorescens* ATCC 39502 strains were grown in Obafluorin Production Medium (OPM) (Yeast extract 0.5%, D-glucose 0.5%, MgSO₄ × 7H₂O 0.1%, and FeSO₄ 0.1%, dissolved in Milli-Q (Merck Millipore) filtered water). A toothpick was used to inoculate 100 ml of OPM seed culture (250 ml Erlenmeyer flask) from a 40% glycerol stock (stored at - 80 °C), with subsequent growth for 24 h at 25 °C, 300 r.p.m. 1 ml of this culture was used to inoculate a 100 ml (500 ml flask) OPM production culture which was incubated under the same conditions for 14 h. Samples were prepared for HPLC/LCMS analyses by extracting 1 ml of culture broth with an equal volume of ethyl acetate by mixing at 1,400 r.p.m. for 15 min. Samples were then centrifuged (1,616*g* for 15 min), and the organic phase was collected and evaporated. The resulting extract was dissolved in MeCN (500 μl) and centrifuged (1,616*g* for 20 min) to remove any remaining cell debris.

**Protein expression and purification.** *E. coli* NiCo21(DE3) (NEB) pLysS strains carrying pET28a(+)-*obaG*, pET28a(+)-*obaH* and pET28a(+)-*obaI* were cultivated in Terrific Broth (TB) at 28 °C and 250 r.p.m. on a rotary shaker until A₆₀₀ₙₘ ∼ 0.5. Protein expression was induced by addition of 0.1 mM IPTG and incubation continued at 18 °C and 200 r.p.m. for 18 h. Cells were pelleted at 2,415*g* and 4 °C, and were subsequently re-suspended in buffer containing 25 mM HEPES (ObaG and ObaH)/50 mM Tris-HCl (ObaI) at pH 7.8 containing NaCl (300 mM), MgCl₂ (15 mM) and glycerol (10%). ObaG and ObaH buffers were further supplemented with PLP (0.4 mM) and ThDP (0.5 mM) respectively. After disruption with an EmulsiFlex-B15 high pressure homogeniser (Avestin, Inc.), cells were pelleted at 26,892*g* and 4 °C for 30 min. The lysed supernatant was incubated with chitin resin with gentle mixing for 30 min to remove any endogenous *E. coli* metal binding proteins. Eluted sample was loaded onto a HisTrap excel (GE Healthcare) Ni-NTA using an ÄKTA pure (GE Healthcare) system. Proteins were washed in 5 CV of their respective buffers containing 10, 30 and 50 mM (His₆-ObaG and His₆-ObaG only) imidazole concentrations. His₆-ObaI was eluted with 20 CV of 50 mM imidazole, and His₆-ObaG and His₆-ObaH were eluted in 20 CV of 250 mM imidazole, all in 2 ml fractions. Oba protein-containing fractions were combined and applied to Amincon columns (30 kDa MWCO) and diluted > 1,000 × to remove imidazole, before being concentrated. The His₆-ObaI sample was further purified by size exclusion over a HiLoad 16/600 Superdex 200 pg column (GE Healthcare). Eluted His₆-ObaI-containing fractions were combined and concentrated in an Amincon column (30 kDa MWCO). Protein samples were stored at 4 °C for *in vitro* assays and long-term storage was at - 80 °C in their respective buffers supplemented with 20% glycerol. Recombinant protein samples were run on SDS–PAGE gels to confirm their size (Supplementary Fig. 18). Protein bands were subsequently cut out, washed, reduced, alkylated and treated with trypsin according to standard procedures adapted from Shevchenko *et al.*[55] The tryptic peptide fragments were analysed by mass spectrometry to further confirm protein identity, using an autoflex Speed MALDI-TOF/TOF mass spectrometer (Bruker Daltonics GmbH).

***In vitro* His₆-ObaG discontinuous L-TTA activity assay.** Reactions were performed in 100 μl reaction volumes comprising 50 μM His₆-ObaG, 10 mM glycine/L-serine/L-threonine or [U-¹³C₄,¹⁵N]L-threonine (98% isotopic purity) (for subsequent NMR experiments) and 10 mM **10**, all in His₆-ObaG buffer (25 mM HEPES pH 7.8, 300 mM NaCl, 15 mM MgCl₂, 0.4 mM PLP). Reactions were initiated by introduction of the enzyme and were incubated at 27 °C and 700 r.p.m. for 2 h. A boiled enzyme sample was used as a negative control. Reactions were terminated by addition of MeOH (100 μl) and were incubated at - 20 °C for 1 h to ensure full enzyme precipitation. Precipitated enzyme was pelleted at 1,616*g* for 30 min before analysis by HPLC. Time course data were similarly collected by terminating the reaction at a range of time points up to 5 h and all time points were assayed in triplicate.

The amenability of His₆-ObaG to alternative substrates was explored by performing the assay described above but using 10 mM of either benzaldehyde or phenylacetaldehyde instead of **10** as a co-substrate with L-threonine. The reverse reaction to generate **10** and L-threonine using **2** with and without acetaldehyde as substrates with His₆-ObaG was also performed as above but using 20 mM concentrations of starting substrates.

Single-substrate kinetic analysis was carried out by performing the L-TTA activity assay with varying concentrations of L-threonine (1–200 mM). Reactions were performed using 25 μM enzyme and were incubated at 27 °C, 700 r.p.m. for 4 min, before quenching, sample processing and HPLC analysis as described previously. Five replicates were performed for each concentration of L-threonine assayed, and data were fitted to the Michaelis-Menten equation using GraphPad Prism 5.04 (GraphPad Software, Inc., La Jolla, USA).

***In vitro* His₆-ObaH decarboxylase activity assay.** Reactions were performed in 100 μl reaction volumes comprising 50 μM His₆-ObaH and 10 mM **9** or phenylpyruvate, all in His₆-ObaH buffer (25 mM HEPES pH 7.8, NaCl (300 mM), MgCl₂ (15 mM) and ThDP (0.5 mM)). Reactions were initiated by introduction of the enzyme and were incubated at 27 °C and 700 r.p.m. for 5 min. A boiled enzyme sample was used as a negative control. Reactions were terminated as described previously. HPLC and LCMS analysis was performed as described for the His₆-ObaG L-TTA activity assay.

**Coupling of His₆-ObaH and His₆-ObaG reactions.** His₆-ObaG was first exchanged into His₆-ObaH buffer (see previous) to avoid reaction of **9** with unbound PLP. Reactions were performed in 100 μL reaction volumes comprising 50 μM His₆-ObaG, 50 μM His₆-ObaG, 10 mM **9** and 10 mM L-threonine, all in His₆-ObaH buffer. Reactions were initiated by introduction of the enzyme and were incubated at 27 °C and 700 r.p.m. for 2 h. Control reactions were also performed in which one or both of His₆-ObaG and His₆-ObaH were boiled before the reaction. Reactions were terminated by addition of MeOH (100 μl) and were incubated at - 20 °C for 1 h to ensure full enzyme precipitation. Precipitated enzyme was pelleted at 1,616*g* for 30 min before analysis by HPLC.

**PLP-dependence of His₆-ObaG.** Recombinant His₆-ObaG was incubated with L-penicillamine[44]. Excess PLP was removed from His₆-ObaG samples by exchanging into a reaction buffer comprising 25 mM HEPES pH 7.8, 300 mM NaCl, and 15 mM MgCl₂, using an Amincon column (30 kDa MWCO). Reactions were performed in 600 μl reaction volumes comprising 20 μM His₆-ObaG and were initiated by addition of L-penicillamine (dissolved in reaction buffer) to a final concentration of 10 mM. Ultraviolet–visible spectra were recorded over 30 min using a Lambda 35 UV/Visible Spectrophotometer (PerkinElmer).

A second spectrophotometric assay was performed using His₆-ObaG samples with excess PLP removed in which protein was treated with NaBH₄ to reduce the His₆-ObaG-PLP aldimine to form an amine adduct[45]. Reactions were performed on ice for 15 min in 600 μl volumes comprising 20 μM His₆-ObaG and were initiated by addition of NaBH₄ (dissolved in reaction buffer) to a final concentration of 1 mM. Samples were analysed by UV/Visible spectrophotometry.

**His₆-ObaI hydroxylamine-trapping assay.** Reaction mixtures comprised 8.5 μM His₆-ObaI, 50 mM Tris-HCl pH 8, ATP (2.25 mM), hydroxylamine (150 mM), amino acid substrate (5 mM), MgCl₂ (15 mM) in a final volume of 300 μl, and were allowed to proceed at 28 °C for 5 h. Boiled enzyme and no substrate reactions were performed as negative controls. Following reaction termination by addition of quenching solution (10% FeCl₃ × 6H₂O and 3.3% trichloroacetic acid made up in 0.7 M HCl), and centrifugation to pellet precipitated protein, samples were transferred to cuvettes and were measured at A₅₄₀ₙₘ on a Spectronic Biomate 3 (Thermo Fisher Scientific).

**Chemical synthesis and isolation of substrates.** For the isolation of **1** and its methanolysis product, the supernatant from 6 l of culture broth was subjected to the liquid–liquid partition using equal volume of ethyl acetate. The organic fraction was concentrated under reduced pressure and the residue dissolved in MeCN. This was subjected to preparative reversed-phase HPLC (C18, 150 × 21.2 mm, 110 Å Phenomenex; A: water; B: MeCN; gradient 0-5 min 10% B (*v/v*), 5-35 min, 10-100% B (*v/v*), 35–40 min 100% B (*v/v*), 40–41 min 100–10% B (*v/v*), 41–45 min, 10% B (*v/v*); flow rate was 20 ml min⁻¹; monitored at 250 nm). The fractions containing **1** and its methanolysis product were combined and further purified by semi-preparative reversed phase HPLC (C18, 150 × 10 mm, 110 Å, Phenomenex; A: water; B: MeCN; isocratic 50% B). **1**: Pale yellow solid, (13 mg). ¹H NMR (400 MHz, MeCN-d₃) δ 8.23 (1H, d, *J* = 8.8 Hz), 8.10 (2H, d, *J* = 8.8 Hz), 7.45 (2H, d, *J* = 8.8 Hz), 7.20 (1H, dd, *J*₁ = 8.2 Hz, *J*₂ = 1.27 Hz), 7.04 (1H, dd, *J*₁ = 7.9 Hz, *J*₂ = 1.2 Hz), 6.82 (1H, t, *J* = 7.96 Hz), 5.75 (1H, dd, *J*₁ = 8.5 Hz, *J*₂ = 6.2 Hz), 5.05 (1H, m), 3.38 (1H, dd, *J*₁ = 15.1 Hz, *J*₂ = 5.1 Hz), 3.21 (1H, dd, *J*₁ = 15.1 Hz, *J*₂ = 5.1 Hz); ¹³C NMR (100 MHz, MeCN-d₃) δ 171.47, 169.11, 150.43, 148.08, 147.02, 145.41, 131.20, 124.64, 120.50, 120.07, 118.68, 114.86, 78.50, 59.87, 36.16 p.p.m. HRMS (*m/z*): [M + H]⁺ calculated for C₁₇H₁₄N₂O₇, 359.0874, found, 359.0872. The NMR data is consistent with published values[35]. Methanolysis product: Pale yellow solid (23 mg). ¹H NMR (400 MHz, MeCN-d₃,) δ 8.12 (1H, d, *J* = 8.8 Hz), 7.60 (1H, d, *J* = 8.6 Hz), 7.48 (2H, d, *J* = 8.7 Hz), 7.27 (1H, d, *J* = 8.4 Hz), 7.02 (1H, d, *J* = 7.7 Hz), 6.80 (1H, d, *J* = 8.1 Hz), 4.77 (1H, dd, *J*₁ = 2.3 Hz, *J*₂ = 8.9 Hz), 4.46 (1H, m), 3.72 (3H, s), 3.00 (1H, dd, *J*₁ = 14 Hz, *J*₂ = 5.2 Hz), 2.93 (1H, dd, *J*₁ = 14 Hz, *J*₂ = 5.2 Hz); ¹³C NMR (100 MHz, MeCN-d₃) δ 171.90, 171.56, 150.51, 148.13, 147.78, 147.17, 131.90, 124.65, 120.34, 120.12, 119.04, 115.69, 72.91, 57.64, 53.58, 41.08 p.p.m. HRMS (*m/z*): calculated for: [M + H]⁺ C₁₈H₁₉N₂O₈, 391.1136, found, 391.1139.

Ethyl 5-(4-nitrobenzyl)-4,5-dihydrooxazole-4-carboxylate (**11**) was synthesized according to a literature procedure[56] using *p*-nitrophenylacetaldehyde (990 mg, 6 mmol) and ethyl isocyanoacetate (750 mg, 6.6 mmol) to yield 910 mg of the racemic *trans*-diastereoisomer. Yield: 55%. $^1$H NMR (400 MHz, CDCl$_3$) $\delta$ 8.08 (2H, d, $J$ = 8.8 Hz), 7.36 (2H, d, $J$ = 8.8 Hz), 6.87 (1H, d, $J$ = 9.4 Hz), 4.72 (1H, dd, $J_1$ = 9.4 Hz, $J_2$ = 2.0 Hz), 4.41-4.36 (1H, m), 4.22–4.13 (2H, m), 2.92-2.78 (2H, m), 1.23 (3H, t, $J$ = 7.2 Hz); $^{13}$C NMR (100 MHz, CDCl$_3$) $\delta$ 170.1, 161.8, 146.8, 145.3, 130.3, 123.6, 72.1, 62.2, 54.6, 40.1, 14.0 p.p.m.; HRMS (*m/z*): [M + H]$^+$ calcd. for: C$_{13}$H$_{15}$N$_2$O$_5$, 279.0975; found, 279.0978.

(2SR,3RS)-2-amino-3-hydroxy-4-(4-nitrophenyl)butanoic acid (**2**) was synthesized according to a literature procedure[56]. Starting from 720 mg of **11** we obtained 570 mg of **2**. Yield: 92%. $^1$H NMR (400 MHz, D$_2$O) $\delta$ 8.31 (2H, d, $J$ = 8.5 Hz), 7.63 (2H, d, $J$ = 8.5 Hz), 4.44 (1H, m), 3.84 (1H, m), 3.27 (1H, dd, $J_1$ = 14.0 Hz, $J_2$ = 3.2 Hz), 3.05 (1H, dd, $J_1$ = 14.0 Hz, $J_2$ = Yield: 92%. 1.6 Hz); $^{13}$C NMR (100 MHz, D$_2$O) $\delta$ 172.5, 146.6, 145.9, 130.5, 123.9, 70.3, 59.2, 39.8 p.p.m. HRMS (*m/z*): [M + H]$^+$ calcd. for: C$_{10}$H$_{13}$N$_2$O$_5$, 241.0819; found: 241.0801. NMR data are in agreement with published values for the (2S,3R)-enantiomer[43].

(2SR,3RS)-2-amino-3-hydroxy-4-(4-aminophenyl)butanoic acid (**8**) was synthesized according to a literature procedure[54]. Starting from 25 mg of **2** we obtained 7 mg of **7** as a yellow solid. $^1$H NMR (400 MHz, D$_2$O) $\delta$ 7.37 (2H, d, $J$ = 8.4 Hz), 7.28 (2H, d, $J$ = 8.1 Hz), 4.39 (m, 1H), 4.05 (1H, d, $J$ = 4.0 Hz), 3.01 (1H, dd, $J_1$ = 14.1 Hz, $J_2$ = 4.3 Hz), 2.82 (1H, dd, $J_1$ = 9.9 Hz, $J_2$ = 14.2 Hz); $^{13}$C NMR (100 MHz, D$_2$O) $\delta$ 170.35, 138.28, 130.86, 128.44, 123.17, 69.71, 57.16, 38.77 p.p.m. HRMS (*m/z*): [M + H]$^+$ calcd. for: C$_{10}$H$_{15}$N$_2$O$_3$, 211.1077 [M + H]$^+$; found, 211.1082.

The enzymatic synthesis of **2** was achieved by scaling up (to 2 ml) the analytical conditions for the ObaG discontinuous assay described above. The reaction was quenched after 2 h by the addition of MeOH (2 ml) and the solution was concentrated under reduced pressure. The resulting crude product was repeatedly subjected to SPE for further purification and to remove excess buffer. Elution with 25% MeOH yielded **2** (1.8 mg, 38%). NMR data were consistent with those of the synthetic reference standard. $[\alpha]_D$ + 48 °, (c = 0.18, H$_2$O) (Literature[43] $[\alpha]_D$ + 50 °, (c = 0.18, H$_2$O)).

**Phylogenetic analyses.** Amino acid sequences of L-TAs and SHMTs were obtained from a previous phylogenetic study[50] and were combined with additional L-TA and SHMT sequences for enzymes which have been characterized or described in the literature, and were obtained from the National Center for Biotechnology Information (NCBI) GenBank database[57] and the Protein Data Bank (PDB)[58]. BLASTP (Basic Local Alignment Search Tool)[59] searches for L-TAs or SHMTs involved in natural product biosynthesis were performed to identify enzymes from these families associated with specialized metabolism. A selection of 10 amino acid sequences for 3-amino-5-hydroxybenzoic acid synthases (AHBASs) described in the literature were obtained to function as an outgroup for phylogenetic analysis as they have been shown to share a recent common ancestor of L-TAs and SHMTs[60] (Source organisms and amino acid sequence GenBank accession numbers used are reported in Supplementary Table 4). Sequences were initially aligned (all alignments performed with default settings) using ClustalX2 (ref. 61), before manual trimming of sequences at *N*- and *C*- termini to remove aberrant sequences (for example, histidine tags) that might interfere with the alignment. Trimmed sequences for Fig. 5 are reported in Supplementary Note 3. Several iterations of alignment and trimming were repeated with different degrees of trimming to ensure that the final tree was relatively consistent and robust (Supplementary Figs 15 and 16). Among the amino acid sequences used, FTase is unique in possessing a C-terminal phosphate-binding domain, and this was also trimmed (Supplementary Fig. 19). MUSCLE[62], in addition to ClustalX2, was trialled for initial alignment before trimming (Supplementary Fig. 10). Trimmed sequences were finally re-aligned with T-Coffee[63], and phylogenetic tree inference was performed using maximum likelihood/rapid bootstrapping under the GTR model using RAxML-HPC BlackBox (8.2.8)[64] via the CIPRES Science Gateway portal[65]. The JTT Protein Substitution Matrix was used and all other parameters were set to default values.

**Data availability.** The authors declare that the data supporting the findings reported in this study are available within the article and the Supplementary Information, or are available from the authors on reasonable request. New nucleotide sequence data have been deposited in NCBI GenBank under the accession codes as follows: **1** BGC (KX931446); **pTS1** (KX931445); **pME3087** (KX931444).

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

## Acknowledgements

This work was supported by the BBSRC via Institute Strategic Programme Grant BB/J004561/1 to the John Innes Centre (JIC). The authors declare no conflicts of interest. We thank Dr Jacob Malone (JIC) for the supply of pME3087 and his invaluable assistance and expertise in *Pseudomonas* genetics. Plasmid pJH10 was a kind gift of Professor Chris Thomas and Dr Joanne Hothersall (University of Birmingham, UK). We also thank Dr Lionel Hill and Dr Gerhard Saalbach (JIC) for their excellent metabolomics support. Genome sequencing was undertaken at the Earlham Institute (Norwich, UK).

## Author contributions

B.W. and T.A.S designed the research. T.A.S., D.H. and B.W. wrote the manuscript. T.A.S. performed detailed bioinformatics and phylogeny analysis, carried out the mutational analysis, purified proteins and ran biochemical experiments. D.H. carried out the ObaG NMR experiments and isolated enzymatically synthesized compounds. D.H. and B.W. synthesized chemical standards. Z.Q. isolated obafluorin and analogues and carried out NMR analysis.

## Additional information

**Competing interests:** The authors declare no competing financial interests.

