## [Peer Review File · Nature Communications]

Reviewers' comments:

Reviewer #1 (Remarks to the Author):

In this manuscript, Scott et al. reported the verification of the gene cluster for obafluorin biosynthesis in *P. fluorescens* and the identification of a transadolase ObaG. Preliminary data suggest that this transadolase ObaG uptakes L-threonine as its substrate and is evolutionarily different comparing to other characterized glycine transadolase. This finding might provide an efficient and useful tool for the synthesis of L-threo-beta-hydroxy-alpha-amino acids, as tested in the manuscript. In general, the quality of the work warrants publication in Nature communication. However, the major issues lie in that the characterization of oba is not quite complete since several pieces of information, as detailed below, is missing. Besides, a more thorough biochemical analysis of ObaG would strengthen the manuscript. I would recommend that the manuscript needs a major revision and hopefully all the comments below are addressed before publication.

Major points:

1. The results presented in Figure 2 are somewhat confusing and misleading.
 - a. For example, the elevation of production of compounds 5 and 6 is not shown in either Figure 2 or SI.
 - b. It will be more straightforward and easier for readers to track the change of mutant profiles if the authors could label peak number above the peak. This type of change should be applied to all other manuscript and SI figures.
 - c. The legend of Figure 2 is confusing too since there is no obaX gene in the locus.
 - d. obaL is involved in the biosynthesis of compound 4, the authors need to explain why supplementing compound 2 to the mutant culture would recovery the production of obafluorin.
 - e. Where is compound 7? This compound was only mentioned in the text but was not shown in any of the figures.
2. Can the authors detect any shunt products or intermediates in obafluorin biosynthesis after genetic mutation? Can all the compounds (1 to 9) shown in Figure 1 be identified using UV or MS? It will be more convincing if the authors could show the complete profile and how the profile changes with genetic mutation instead of showing each individual UV peak. Right now they are only showing the loss of compound 1 in Figure 2.
3. The authors should also show the boundary of the oba cluster and if there is any data (bioinformatic or experimental) which define the boundary of the cluster, they should show it in the manuscript or SI. What are the surrounding genes? Are these involved or no likely to be involved in obafluorin biosynthesis? If not, what are their proposed functions?
4. The in vitro reconstitution of ObaH, ObaG, and Obal strengthens the paper.

a. It will make the results more complete if the authors could supply the kinetic data of the enzymes ObaH and ObaG. Especially for the transadolase ObaG, how long does the enzyme to convert its substrate? Any time course related data?

b. Did the authors give it a try to in vitro reconstitute the NRPS ObaI? Could the final product 1 be detected?

5. Again, the authors need to be very careful with chemicals and their corresponding numbers. For exp, in SI Figure 3, 2,3-DHBA is labeled as compound 2 which is compound 4 in the manuscript.

Minor points:

1. The authors might want to briefly cover some other BGCs that are involved in beta-lactone natural product biosynthesis. This would help the reader to understand how novel the oba operon is and how different it is comparing to other beta-lactone BGCs.

2. The production yield of ObaG is 57%. How is this yield calculated? By calculating the loss of the starting material? Or by calculating the accumulation of the final products.

3. Page 5, should read "of which".

Reviewer #2 (Remarks to the Author):

As I started reading this manuscript, I was concerned that the roles of threonine aldolases have been observed and studied in the past. These papers are referenced, notably the work of van Lanen, and, altho the "sample" is small, the work is good. One wonders if this ms. is sufficiently novel to merit publication in a Nature journal. The opinions of the other reviewers are of interest here. However, as the narrative progresses, a fuller discussion ensues and the role of this aldolase is placed in a larger context of obafluorin biosynthesis, whose BGC is also described. In the end for this reviewer the sum of the data and discussion merits a Nat. Chem. report.

I have a few substantive comments and several stylistic ones.

The first is on p. 4 in which bioinformatic evidence "allowed us to identify...all of the biosynthetic activities required for biosynthesis of 1." Is this not too strong? Bioinformatics doesn't prove; it guides/suggests/supports, but experiments are needed. Perhaps "...all of the activities that appeared necessary for..."

Second, on p. 6 and again later it is stated: "MbtH proteins are required for the activity of NRPS A-domains.." This statement is incorrect. Some A-domains do not require these auxiliary proteins for in vitro activity (and presumably not for in vivo activity either), but some do. Baltz has a review *J Ind Microbiol Biotechnol* (2011) 38:1747-1760 and other examples if you search forward. There are a small number of MbtH domains in cis that are just upstream of A domains. In vitro experiments are lacking (and the experiments were not done here) to test if such A domains REQUIRE the adjacent MbtH to function, i.e. undergo the classical PPI exchange assay. In any event, I don't think it right (and is not explicitly demonstrated) that either or both A-domains "require" the MbtH to function. The entirety of p. 6 needs to be rethought and rewritten.

Third, and further to this point, some evidence (good) was obtained that A1 activates 2,3-diOH-benzoate and A2 "threonine", actually key intermediate 2. I would have preferred individual domain assays \pm MbtH, but the whole NRPS was expressed and A-domain substrate selection was tested--rather few substrates and not super specific. The later was picked up upon and the expected suggestions about engineering possibilities raised. This topic seems to be work for the future. That's fine. I accept that bioinformatics gave predictions as to what is recognized by A1 and A2, which were backed up by experiment for the protein as a whole. Not mentioned is the recent finding (Sussmuth CCB 2016, Meyer et al. and more coming by others) that A-domain context can sometimes affect outcomes of substrate selectivity experiments. That the whole NRPS was used gives confidence that the observations reflect the true selection of A1 and A2.

Fourth, the several, mostly failed, experiments with nitroaryl intermediates and shunt products are interesting and the authors are congratulated for following up their findings. I suspect their rationale for cellular behavior is correct.

Fifth, does beta-lactone formation want to be addressed, or is this the subject of a subsequent paper?

Sixth, is an observation. The NRPS domain architecture is quite unusual and interesting. The central TE and 'front to end" setting of the MbtH-like domain and A1 at the C-terminus is unexpected and interesting that the system works. As noted above, it is not shown how important the MbtH is to a.a. substrate activation (my one suggestion to improve the ms,) but a sufficient body of results have been gathered to generate a pretty convincing story.

Now to stylistic points:

1, on p. 2 I see "due to issues with" and later on p. 11 "due to Issues of low" I bristle at this colloquialism in a scientific paper...sort of like nouns becoming verbs like "tasked" or "sourced"

2, In multiple places are dangling "this" phrases; e.g. p.3 "Despite this, ...", p. 4 "This allowed...", p. 5 "Consistent with this,", p. 7/8 "Consistent with this,...", and so forth.

3, p. 7, line 6: replace text with "...but neither 2 nor 4 was accumulated".

4, p. 8, half way down, dependence is misspelled

5, p. 11, 8 lines from bottom: 'cleave' should be 'cleavage'

Reviewer #3 (Remarks to the Author):

This is an interesting paper describing some biosynthetically novel steps in the biosynthesis of the beta-lactone metabolite obafluorin. The non-standard architecture of the NRPS and the transaldolase reaction involved in formation of intermediate 2 will be of significant interest. The paper itself is not that easy to read and I feel a more concise and more logically ordered manuscript would be more acceptable.

Phrases like 1 biosynthesis and 4 production are very stilted and should be replaced by biosynthesis of and production of 1/4.

Some other comments –

6 is presumably NOT a shunt product from 9. Should 6 not be the phenylacetate rather than the phenylpropionate shown?

On page 5 "of-which" should be of which.

Figure 2B does not really show loss of production of 1, 5 and 6.

Further down page 5 ... mutants of 4 biosynthesis .. does not make sense to me.

obaC should be ObaC

Page 6 - what are MbtH proteins , what do they do and why are they required?

ObaK is described as an Aryl carrier protein. Could the authors elaborate on this?

At the bottom - what is meant by the "product" of obaK. Does this mean the acylated ACP?

Again foot of page 7, I cannot readily see how 6 can be a product of detoxification of 9.

Re. NCOMMS-16-23999 revisions

The specific points raised by each referee (shown in red) are addressed below:

Referee 1

In this manuscript, Scott et al. reported the verification of the gene cluster for obafluorin biosynthesis in *P. fluorescens* and the identification of a transadolase ObaG. Preliminary data suggest that this transadolase ObaG uptakes L-threonine as its substrate and is evolutionarily different comparing to other characterized glycine transadolase. This finding might provide an efficient and useful tool for the synthesis of L-threo-beta-hydroxy-alpha-amino acids, as tested in the manuscript. In general, the quality of the work warrants publication in Nature communication. However, the major issues lie in that the characterization of oba is not quite complete since several pieces of information, as detailed below, is missing. Besides, a more thorough biochemical analysis of ObaG would strengthen the manuscript. I would recommend that the manuscript needs a major revision and hopefully all the comments below are addressed before publication.

As described above we have performed a range of biochemical experiments on ObaG which we hope is now more than sufficient to address this issue.

Major points:

1. The results presented in Figure 2 are somewhat confusing and misleading.

a. For example, the elevation of production of compounds 5 and 6 is not shown in either Figure 2 or SI.

b. It will be more straightforward and easier for readers to track the change of mutant profiles if the authors could label peak number above the peak. This type of change should be applied to all other manuscript and SI figures.

Ia&b: the elevated production levels for compounds 5 and 6 (now 6 and 7) are shown in Fig 2 and in the ESI figures, but we understand that they were not labelled to make this clear on the chromatograms. To address this point we have labelled the compound peaks on the chromatograms in the main manuscript and ESI as requested in point 1b.

c. The legend of Figure 2 is confusing too since there is no obaX gene in the locus.

Ic: we agree that the legends of figures containing HPLC traces may be confusing and have therefore altered this and labelled each of the chromatograms to aid with clarity.

d. obaL is involved in the biosynthesis of compound 4, the authors need to explain why supplementing compound 2 to the mutant culture would recovery the production of obafluorin.

Id: although *obaL* is required for making compound **4** (now **5**), a possible consequence of this was accumulation of the other key pathway intermediate (compound **2**), which, like **5**, is an anticipated substrate for the NRPS. This has been clarified in the text.

e. Where is compound 7? This compound was only mentioned in the text but was not shown in any of the figures.

1e. we discuss compound **7** (now **8**) as a theoretical intermediate on the way to making compound **2**, and therefore tested for its accumulation in various mutants as a synthetic standard was available. For clarity and consistency this and the structure of L-4- aminophenylalanine (now introduced as **3**) have been added to the bottom of Figure 1 and noted in the legend.

2. Can the authors detect any shunt products or intermediates in obafluorin biosynthesis after genetic mutation? Can all the compounds (1 to 9) shown in Figure 1 be identified using UV or MS? It will be more convincing if the authors could show the complete profile and how the profile changes with genetic mutation instead of showing each individual UV peak. Right now they are only showing the loss of compound 1 in Figure 2.

2. as described in the response to point 1a, the shunt metabolites **5** and **6** (now **6** and **7**) are clearly visible in the relevant figures and have now been annotated, as have other relevant compounds. As stressed throughout the manuscript, while these intermediates are accumulated in several mutants, and their presence or lack thereof becomes diagnostic of certain biosynthetic steps, we were not able to observe accumulation of the key intermediates **2** and **5** (was **4**), or most of the others (see above). Synthetic standards were available for all compounds other than what is now compound **4** (was **3**) and we tested for **8** (was **7**). As discussed in the manuscript we suggest that our data suggest obafluorin biosynthesis is carefully regulated to avoid the accumulation of toxic intermediates such as **5** and **10** (were **4** and **9**), and has pathways such as those leading to the shunt metabolites **6** and **7** which can detoxify the aldehyde **10**.

Please note, that as these compounds (other than what is now **4**) were all available we also always examined their ability to chemically complement blocked mutants where appropriate.

As can be seen in the ESI figures - where we have more space - that the full chromatograms are shown for certain of the mutants and verify that the regions we show in the main document figures are the critical ones. As noted above and throughout the manuscript we searched all experiments exhaustively for the accumulation of intermediates and shunt metabolites.

3. The authors should also show the boundary of the oba cluster and if there is any data (bioinformatic or experimental) which define the boundary of the cluster, they should show it in the manuscript or SI. What are the surrounding genes? Are these involved or no likely to be involved in obafluorin biosynthesis? If not, what are their proposed functions?

3. we have added additional data to the ESI Table S1 which show the genes surrounding the *oba* BGC. No necessary biosynthetic function could be ascribed to any of these genes, and, given the mutagenesis data which identified all of the activities required for the biosynthesis of obafluorin, we did not see the need for their additional mutational analysis.

Despite this, and as discussed in the Conclusions section, we did delete the four genes (*orfs2-5*) which comprise an apparent glycine cleavage system as it was possible they were involved in L-threonine supply. However, their deletion had no effect on obafluorin production. This is discussed in the main manuscript.

4. The in vitro reconstitution of ObaH, ObaG, and Obal strengthens the paper.

a. It will make the results more complete if the authors could supply the kinetic data of the

enzymes ObaH and ObaG. Especially for the transaldolase ObaG, how long does the enzyme to convert its substrate? Any time course related data?

4a. as described above we have performed additional biochemical analysis of ObaG including preliminary kinetic analysis. As part of these studies we set up a coupled ObaH- ObaG assay for the one pot conversion of compound **9** (was **8**) and L-threonine into **2**.

b. Did the authors give it a try to in vitro reconstitute the NRPS ObaI? Could the final product **1** be detected?

4b. we did not attempt to reconstitute the NRPS ObaI due to technical difficulties. This work will form part of further studies and publications regarding obafluorin biosynthesis – see below in response to Referee 2.

5. Again, the authors need to be very careful with chemicals and their corresponding numbers. For exp, in SI Figure 3, 2,3-DHBA is labeled as compound **2** which is compound **4** in the manuscript.

5. we corrected the error regarding numbering of compound **4** (now **5**) in the ESI figure, and have carefully checked the manuscript for further such errors.

Minor Points:

1. The authors might want to briefly cover some other BGCs that are involved in beta-lactone natural product biosynthesis. This would help the reader to understand how novel the oba operon is and how different it is comparing to other beta-lactone BGCs.

1. we chose not to enter a discussion of the BGCs and biosynthesis of other beta-lactone natural products as we felt this would be distracting from the flow of the manuscript. A range of beta-lactone natural products are described that the reader can follow up to investigate this question if desired.

2. The production yield of ObaG is 57%. How is this yield calculated? By calculating the loss of the starting material? Or by calculating the accumulation of the final products.

2. The production yields for ObaG were calculated using calibrations for authentic material using both disappearance of product and appearance of substrate. We described data for the accumulation of product but this is in agreement with the data for the depletion of substrate (this was checked) and for the time course data in Figure S12a we show both.

3. Page 5, should read “of which”.

3. the text has been corrected

Referee 2:

As I started reading this manuscript, I was concerned that the roles of threonine aldolases have been observed and studied in the past. These papers are referenced, notably the work of van Lanen, and, altho the "sample" is small, the work is good. One wonders if this ms. is sufficiently novel to merit publication in a Nature journal. The opinions of the other reviewers are of interest here. However, as the narrative progresses, a fuller discussion ensues and the role of this aldolase is placed in a larger context of obafluorin biosynthesis, whose BGC is also described. In the end for this reviewer the sum of the data and discussion merits a Nat. Chem. report.

I have a few substantive comments and several stylistic ones.

The first is on p. 4 in which bioinformatic evidence "allowed us to identify...all of the biosynthetic activities required for biosynthesis of 1." Is this not too strong? Bioinformatics

doesn't prove; it guides/suggests/supports, but experiments are needed. Perhaps "...all of the activities that appeared necessary for..."

1. We have altered the language on p.4 as requested (regarding the use of language to describe the utility of bioinformatics data).

Second, on p. 6 and again later it is stated: "MbtH proteins are required for the activity of NRPS A-domains.." This statement is incorrect. Some A-domains do not require these auxiliary proteins for *in vitro* activity (and presumably not for *in vivo* activity either), but some do. Baltz has a review *J Ind Microbiol Biotechnol* (2011) 38:1747-1760 and other examples if you search forward. There are a small number of MbtH domains in *cis* that are just upstream of A domains. *In vitro* experiments are lacking (and the experiments were not done here) to test if such A domains REQUIRE the adjacent MbtH to function, i.e. undergo the classical PPI exchange assay. In any event, I don't think it right (and is not explicitly demonstrated) that either or both A-domains "require" the MbtH to function. The entirety of p. 6 needs to be rethought and rewritten.

2. We agree that MbtH-like proteins are not required for the function of every adenylation (A)-domain and have amended the text appropriately, and added additional references.

Third, and further to this point, some evidence (good) was obtained that A1 activates 2,3-diOH-benzoate and A2 "threonine", actually key intermediate 2. I would have preferred individual domain assays \pm MbtH, but the whole NRPS was expressed and A-domain substrate selection was tested--rather few substrates and not super specific. The later was picked up upon and the expected suggestions about engineering possibilities raised. This topic seems to be work for the future. That's fine. I accept that bioinformatics gave predictions as to what is recognized by A1 and A2, which were backed up by experiment for the protein as a whole. Not mentioned is the recent finding (Sussmuth CCB 2016, Meyer et al. and more coming by others) that A-domain context can sometimes affect outcomes of substrate selectivity experiments. That the whole NRPS was used gives confidence that the observations reflect the true selection of A1 and A2.

3. We have amended the text regarding the substrate specificity of A-domains and appreciate the Referees comments on this.

Fourth, the several, mostly failed, experiments with nitroaryl intermediates and shunt products are interesting and the authors are congratulated for following up their findings. I suspect their rationale for cellular behavior is correct.

4. We appreciate the comments regarding our efforts to probe the accumulation of intermediates/feeding of intermediates to blocked mutants, and our analysis of the nitroaryl shunts 5 and 6 (now 6 and 7) which became diagnostic *in vivo* markers of certain biosynthetic steps. As noted in the response to Referee 1 above, as we had syntheti standers for all but one compound we were able to check for their presence in all experiments. This was frustrating work but we thought it important to use synthetic standards to verify the *in vivo* phenotypes.

Fifth, does beta-lactone formation want to be addressed, or is this the subject of a subsequent paper?

5. Beta-lactone formation by ObaI will be the focus of future work.

Sixth, is an observation. The NRPS domain architecture is quite unusual and interesting. The central TE and 'front to end" setting of the MbtH-like domain and A1 at the C-terminus is unexpected and interesting that the system works. As noted above, it is not shown how important the MbtH is to a.a. substrate activation (my one suggestion to improve the ms,) but a sufficient body of results have been gathered to generate a pretty convincing story.

6. While a discussion of the relevance of MtbH-like domains has been addressed above, we agree that the ObaI architecture is extremely interesting and future experiments will address this.

Stylistic points:

1, on p. 2 I see "due to issues with" and later on p. 11 "due to issues of low" I bristle at this colloquialism in a scientific paper...sort of like nouns becoming verbs like "tasked" or "sourced"

1. We have addressed 'due to issues with' statements as requested.

2, In multiple places are dangling "this" phrases; e.g. p.3 "Despite this, ...", p. 4 "This allowed...", p. 5 "Consistent with this, ...", p. 7/8 "Consistent with this,...", and so forth.

2. We have addressed the dangling 'this' phrases as requested.

3, p. 7, line 6: replace text with "...but neither 2 nor 4 was accumulated".

3. Sentence has been changed (as has much of this section to improve clarity).

4, p. 8, half way down, dependence is misspelled

4. Section has been changed so spelling altered.

5, p. 11, 8 lines from bottom: 'cleave' should be 'cleavage'

5. Spelling has been corrected

Referee 3:

This is an interesting paper describing some biosynthetically novel steps in the biosynthesis of the beta-lactone metabolite obafluorin. The non-standard architecture of the NRPS and the transaldolase reaction involved in formation of intermediate 2 will be of significant interest. The paper itself is not that easy to read and I feel a more concise and more logically ordered manuscript would be more acceptable.

- We have modified several sections of the manuscript, in particular the results and discussion section, in order to make it clearer and more logical.

Phrases like 1 biosynthesis and 4 production are very stilted and should be replaced by biosynthesis of and production of 1/4.

- The stilted phrases highlighted by this referee have been changed throughout the manuscript.

6 is presumably NOT a shunt product from 9. Should 6 not be the phenylacetate rather than the phenylpropionate shown?

- We agree that the structure of 6 (now 7) was drawn incorrectly and should be phenylacetate – this has been corrected in Figure 1.

On page 5 "of-which" should be of which.

- The 'of-which' issue has been corrected on page 5.

Figure 2B does not really show loss of production of 1, 5 and 6.

- Figure 2B does clearly show the loss of production of 1, 5 (6) and 6 (7) but has not been labelled appropriately – see response to Referee 1 above – this has now been improved by better labelling of the compound peaks and the individual chromatograms.

Further down page 5 ... mutants of 4 biosynthesis .. does not make sense to me.

- The nonsense statement on page 5 has been corrected – as already noted this section has been extensively rewritten to improve its clarity.

obaC should be ObaC

- We have corrected obaC to ObaC and any other such typos where we can find them.

Page 6 - what are MbtH proteins , what do they do and why are they required?

- As noted above for Referees 1 and 2 our explanation of the role of MtbH-like domains has been significantly enhanced.

ObaK is described as an Aryl carrier protein. Could the authors elaborate on this?

- The role of ObaK has been explained in more detail within the text.

At the bottom - what is meant by the "product" of obaK. Does this mean the acylated ACP?

- We have clarified what the ‘product’ of ObaK means as instructed.

Again foot of page 7, I cannot readily see how 6 can be a product of detoxification of 9.

- As noted above, the structure of **6** (now **7**) has been corrected in Figure 1.

Please note there is also a revised ESI document which contains new information.

Reviewers' Comments:

Reviewer #1:

Remarks to the Author:

All the comments and suggestions I mentioned in the review have been taken care of and I would recommend publishing the manuscript as it is.

Reviewer #2:

Remarks to the Author:

This is a much improved ms. that thoughtfully addresses the Reviewers' main substantive criticisms. The writing has improved dramatically and the flow of the argument, augmented by new and additional experimental data, combine to give now a solid contribution to NChem. I recommend acceptance w/o further change.

Reviewer #3:

Remarks to the Author:

The authors have responded satisfactorily to the points raised in the original review. The work will be of general interest and I am happy for it to be accepted as is.